# Propagation of an Epigenetic Age-Related Disorder in Almond Is Governed by Vegetative Bud Ontogeny Rather Than Chimera-Type Cell Lineage

**Thomas M. Gradziel * and Kenneth A. Shackel**

Department of Plant Sciences, University of California, One Shields Avenue, Davis, CA 95616, USA; kashackel@ucdavis.edu
* Correspondence: tmgradziel@ucdavis.edu

**Abstract:** Almond (*Prunus dulcis* [Mill.] D.A. Webb) represents a model system for the study of epigenetic age-related disorders in perennial plants because the economically important noninfectious bud-failure disorder is well characterized and shown to be associated with the clonal-age of the propagation source. Epigenetic changes regulating disorders such as changes in methylation or telomere-length shortening would be expected to occur in shoot apical meristem initial cells since subsequent daughter cells including those in ensuing shoot axillary meristems show an irreversible advance in epigenetic aging. Because multiple initial cells are involved in meristem development and growth, such 'mutations' would be expected to occur in some initial cells but not others, resulting in mericlinal or sectorial chimeras during subsequent shoot development that, in turn, would differentially affect vegetative buds present in the leaf axils of the shoot. To test this developmental pattern, 2180 trees propagated from axillary buds of known position within asymptomatic noninfectious bud-failure budstick sources were evaluated for the disorder. Results demonstrate that relative bud position was not a determinant of successful trait propagation, but rather all axillary buds within individual shoots showed very similar degrees of noninfectious bud-failure. Control is thus more analogous to tissue-wide imprinting rather than being restricted to discrete cell lineages as would be predicted by standard meristem cell fate-mapping.

**Keywords:** somaclonal; cell-fate; shoot-apical-meristem; aging



## 1. Introduction

Genetic mosaics appear to be common among long-lived tree species [1–5]. Zahradníková et al. [5] found age-related mosaicism in most long-lived angiosperm tree species examined and characterized their developmental patterns based on resultant sectorial and mericlinal chimeras. Sarkar et al. [2] and later Plomion et al. [3], who studied 234 and 100-year-old pedunculate oak trees (*Quercus robur*), respectively, identified multiple variants, whose sequential appearance in the tree could be traced along nested chimera sectors in progressively older branches. Recently, epigenetic mosaics have been shown to be present in a sectorial chimera of Japanese apricot (*Prunus mume*), where methylation differences in the stem apical meristem (SAM) were found to be associated with changes in blossom color [6]. Lynch [7] estimated that approximately 1.61 mutations occur per base per cell division in *Arabidopsis*, which would correspond to approximately one mutation every six cell divisions. Klekowski and Godfrey [8] had previously reported that mutation rates in some long-lived trees were 25 times higher than those reported in annual plants. The majority of these mutations remain undiscovered because most plant cells are non-dividing, limiting subsequent tissue expression of altered genetics and/or epigenetics. Opportunities to study such changes primarily occur with budsports, where individual buds from the parent plant show novel and often propagatable phenotypes. In budsports, the change occurs in SAM cells where subsequent divisions produce plant tissues in which

altered expression and development can be differentiated and studied within the resultant shoot sectors of primarily anticlinally dividing daughter cells [9]. The angiosperm SAM has a tunica-corpus structure containing multiple cell initials so that mutation of any individual initial cell within the meristem would be transmitted to daughter cells and subsequent shoot tissues as distinct sectors. As budsports are an important source for new cultivars, their origin and management are well-characterized [9,10] with management typically involving recurrent selections of axillary buds located within chimera sectors of affected shoots until stable propagation of the novel phenotype is achieved [11]. Careful selection of propagation bud source is required to stably propagate desirable sports such as changes in apple skin color [12] or grapefruit pulp color [13] and to avoid disorders such as rootstock collapse in pistachio [14], nonproductive clones of cherry [15] and almonds [16], and clone-decline in winegrape [17]. Noninfectious bud-failure (NBF) is an economically important epigenetic disorder in almond resulting in symptoms of shoot failure with subsequent yield loss [16,18]. NBF has been shown to be transmitted to both vegetative propagules and sexual progeny without a pathogenic origin or epidemiological pattern of dispersion [19]. Epigenetic-like rather than genetic control has been demonstrated by the age-dependent change in phenotypic expression within the same genotype (clone) [16,20–22]. Epigenetic mechanisms including DNA methylation have been implicated in phenotypic variation in several plant species [23,24] including almond [25,26]. Recently, Fresnedo-Ramírez et al. [26] demonstrated that that NBF-exhibition is associated with DNA methylation and associated chronological age in almond.

NBF remains a serious threat to the continued production of the dominant cv. Nonpareil that is currently planted on more than 0.6 million acres and has resulted in dramatic reductions in plantings of cv. Carmel, which was the second most important commercial cultivar in the early 2000s before being afflicted by NBF [20–22]. Because NBF is a clone-age related disorder, control in 'Nonpareil' has been achieved through the careful selection, maintenance, and propagation of low clone-age propagation sources [22]. Similar propagation source analysis for the cv. Carmel also demonstrated an increase in NBF expression with increased age of clone propagation-sources [20]. Results showed that careful selection of budwood source used for tree propagation provided the most effective control of subsequent NBF expression in ensuing commercially grown trees. Trees propagated from the original 'Carmel' breeder's seedling tree showed the lowest proportion of NBF with increased NBF observed with increased generations of propagation (where generation refers to the number of propagation cycles from the original seedling tree). Propagation methods to minimize NBF expression in commercial tree clones is thus similar to bud-propagation practices to avoid undesirable genetic chimeras. However, the relative importance of individual propagation bud-source at different positions within the donor shoot has not previously been evaluated. This study reanalyzes the Kester et al. [20] results to study differences in final NBF expression of trees budded with axillary buds from different positions within the budstick to test the chimera model for vegetative transmission of epigenetic-like mosaics in perennial plants.

## 2. Materials and Methods

The 'Carmel' almond (*Prunus dulcis* L.) propagation sources tested were those used by commercial nurseries during the peak of 'Carmel' plantings in the early 1990s. A total of 62 asymptomatic source trees from 11 different nurseries were evaluated. For each single-tree source, separate budsticks were collected in May from the four quadrants of the tree (roughly northwest, northeast. southwest, southeast). Buds from each budstick were single June-budded [27]) to 'Nemaguard' rootstocks, maintaining the original order on the budstick in the nursery row. After shoots had grown out from each bud, each tree was labeled with a unique number identifying nursery, individual tree, individual branch, individual budstick, and position of individual bud on the budstick. This identification number remained with each tree through the remaining multi-year evaluation. These vegetative progeny trees were then planted in replicated blocks and grown in a commercial

orchard in Lost Hills, California in Kern Co. in the western San Joaquin Valley (35.6894 N, −119.7555 W) where high summer temperatures favor the expression of NBF symptoms. NBF in individual progeny trees was rated as previously described in Kester et al. [20] using visual ratings based on the occurrence of failed buds on current season growth: 0 = no NBF; 1 = NBF expression limited to 1 tree scaffold; 2 = NBF present on more than 1 but less than half of total tree scaffolds; 3 = NBF present on the majority of scaffolds; 4 = NBF present on all scaffolds; and 5 = severe NBF present on all scaffolds with tree in decline. The entire block of 2180 individual trees were rated annually in March over a seven-year test period by at least three trained evaluators, who reached consensus on each rating.

*Statistical Methods*

A variance component analysis (maximum likelihood estimates) was performed on NBF scores at the end of the 7-year experiment using SAS V9.4 PROC VARCOMP (SAS Institute, Cary, NC, USA). An overall analysis was based on estimating the variance due to nurseries, propagation source trees within nurseries, and sticks within source trees. A specific variance due to sticks and buds was separately estimated by performing a separate analysis for each source tree, assuming that the error term from this analysis would largely represent variation due to buds. For this analysis, the average NBF score for each propagation source tree was used as a measure of NBF potential for that propagation source tree.

## 3. Results and Discussion

All axillary buds from the same budstick developed very similar levels of NBF in ensuing vegetatively propagated progeny trees irrespective of position on the budstick (Table 1, Figure 1). Leaves (and associated axillary buds) are developmentally separated by approximately 120° in the almond SAM, resulting in a phyllotaxy where every third leaf/axillary-bud would be roughly aligned with the same SAM derived shoot daughter-cell sectors [28]. Consequently, any NBF-expressing sectors predicted by the chimera model for vegetative transmission of budsport mosaics should result in a general sector overlap and so altered expression at roughly every third axillary bud with intervening buds remaining unaffected (depending upon the size of the presumed sectorial chimera). However, neither this pattern nor high within-budstick bud-score variability was observed even when 10 or more consecutive axillary buds were evaluated. As with the original study [20], the nursery source and propagation source-tree within the nursery showed the strongest association with subsequent NBF expression. Budstick within propagation source trees, and bud position within the budsticks proved to be poor predictors of NBF expression in subsequent vegetative progeny trees. The importance of nursery propagation source is further demonstrated in Figure 2 where average NBF expression is plotted for each nursery source and where the developmental history or propagation lineage of these different nursery propagation sources was projected based on available propagation records. Propagation source A represents the original seedling 'Carmel' tree with source B representing an early commercial orchard from which budwood for the next generation orchards (C, D, E, and F) was collected. Commercial nurseries tend to use more recently planted orchards as a source of new budwood because the greater growth of young trees results in longer budsticks with more viable buds per budstick; both of which are desirable for propagation. Increasing NBF progression is associated with increased cycles of propagation from the original seedling tree. Increasing NBF progression is also associated with increased age/development within individual trees [18]. For each propagation source, the final average NBF severity was closely related to the time of initial NBF expression in orchard propagated trees from that source [20]. Even trees propagated from the original 'Carmel' seedling tree showed NBF by year 10 while trees propagated from high NBF sources and grown in high temperature environments conducive to NBF expression showed complete penetrance by year 8 (i.e., terminal growth on all scaffolds

expressed NBF symptoms). In contrast, commercial trees propagated from cv. Nonpareil low-NBF sources have shown no NBF expression after 20 years of commercial production under strongly NBF inducing environments [21]. Interestingly, these low-NBF propagation sources originated by pushing basal epicormic buds from almost century-old 'Nonpareil' trees (i.e., pushing basal dormant buds originally laid down in young trees only a few propagation generations removed from the original 'Nonpareil' seedling tree, which was released in 1879). Consequently, the 'Carmel' almond cultivar represents a unique opportunity to study age-related disorders in perennial crops since the full incremental aging process from original symptomless seedling tree to complete NBF expression is present in different propagation-sources and, at even finer increments, within individual trees of the different propagation-sources.

　　All axillary buds within a given 'Carmel' shoot appear to be similarly effective in transmitting the altered NBF phenotype, in marked contrast to the expected variability predicted by the chimera model. These results suggest that a different mechanism is involved, more analogous to tissue 'imprinting' as seen with flower induction or the phase change from juvenile to mature forms in plants. However, unlike the more quantitative aspect of flower induction, NBF appears qualitative and irreversible since no reversions have been detected in the thousands of trees evaluated over multiple decades [21]. Thomas [29] reported a similar situation for tree species of Quercus, Fagus, and Carpinus, where the abscission of all leaves borne on lower, juvenile-stage shoots is suppressed with desiccated leaves remaining attached until being displaced by new leaves in the spring. In contrast, all leaves on adult shoots are uniformly shed when senescent in autumn. He further concluded that this change from juvenile to mature was not related to plant size, proximity to the root, or number of dormancy-growth cycles, but was an intrinsic property of dividing cells at the shoot apex, as initially proposed by Robinson and Wareing [30]. While most studies of the juvenile to mature phase change in plants support genetic and epigenetic control through complex internal and external signaling and regulatory feedback pathways [30–32], a recent study by Ahsan et al. [33] reports that the juvenile to adult phase transition in the tree crops avocado, mango, and macadamia was closely related to the sequential activity of two micro RNAs in leaves supporting an earlier proposal by Wang et al. [34] for using epigenetic markers based on micro RNAs as a diagnostic for juvenility. However, because leaf drop is associated with growth cycles in tree species, it is not clear whether micro-RNA activity was a cause or consequence of the phase transition. Similarly, Fresnedo-Ramírez et al. [26] demonstrated that DNA-(de)methylation status was associated with both clone age as well as level of NBF exhibition in almond, but it could not be determined whether the most promising epigenetic markers were directly diagnostic for NBF or, like NBF, were merely associated with clone-age within the limited propagation sources tested.

**Table 1.** Maximum likelihood estimates for the variance components of final noninfectious bud-failure (NBF) in 'Carmel' almond expression in the overall dataset.

| NBF Variance Due to | Variance |
|---|---|
| Nursery source | 0.838 |
| Propagation source tree with the nursery | 1.25 |
| Budstick within propagation source tree | 0.448 |
| Error (Bud position within budstick) | 0.477 |

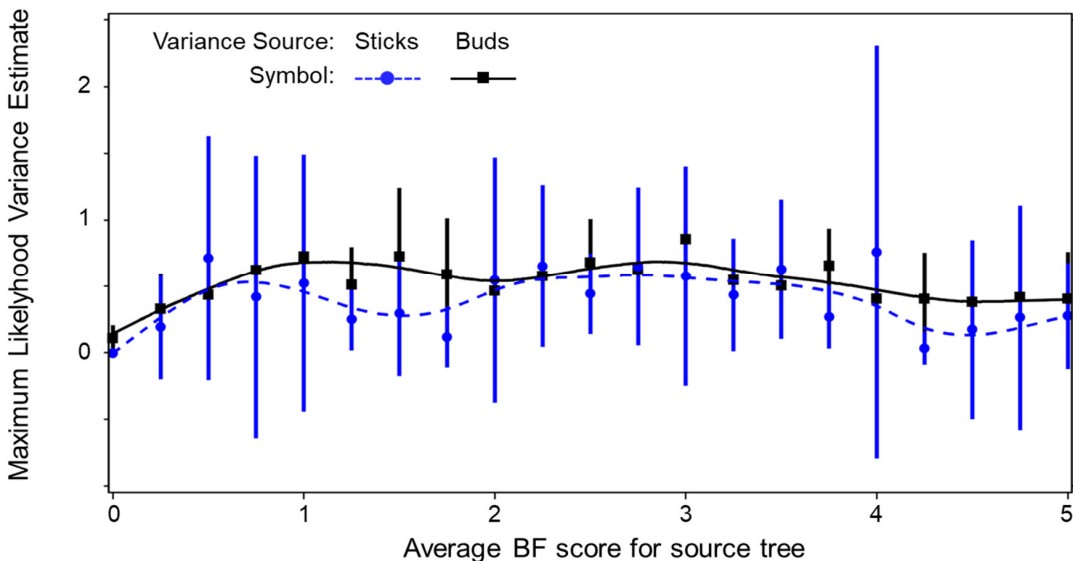

**Figure 1.** Variance due to sticks and buds of 'Carmel' almond as estimated by analyzing each propagation source tree, assuming that the error term would largely represent variation due to buds, and where average BF (noninfectious bud-failure) score for each propagation source tree is used as a measure of the NBF potential for that propagation source tree.

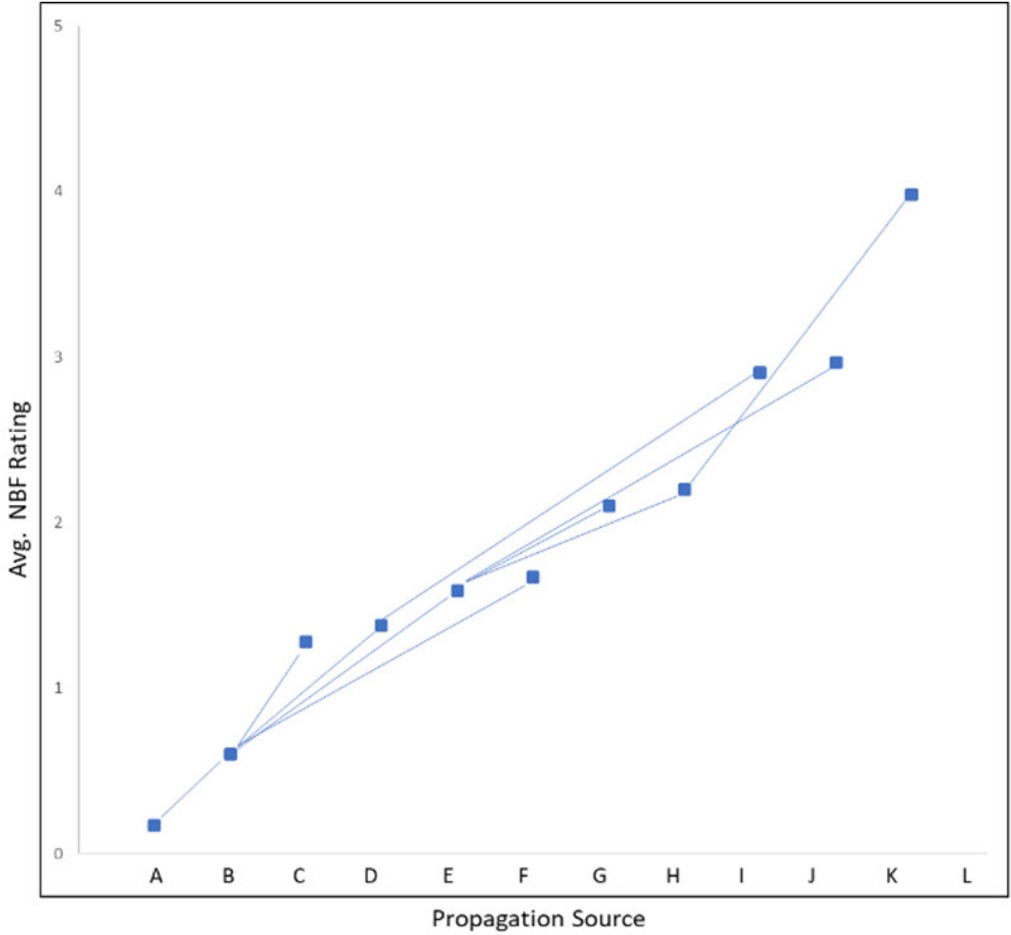

**Figure 2.** Average noninfectious bud-failure (NBF) expression for the different nursery propagation sources of 'Carmel' almond with the relatedness or dissemination lineage of propagation sources plotted based on available propagation records. (The letter A identifies the original breeder seedling tree while B through L identify different commercial nursery propagation sources).

## 4. Conclusions

Novel phenotypes resulting from budsport-type mosaics in perennial plants have been shown to be the result of both genetic [35] as well as epigenetic [6] changes. Management of these changes is primarily through the careful selection of propagation sources based on current generation and/or next generation phenotyping. Efforts to develop more precise molecular markers for the underlying genetic/epigenetic control follow similar selection strategies, typically using the standard chimera model of a distinct origin in SAM cell initials, resulting in distinct shoot sectors derived from the predominantly anticlinal division of daughter cells. In the relatively well studied NBF epigenetic disorder, however, these assumptions of discrete epigenetic sectors have not been supported in large-scale vegetative progeny testing. A more detailed knowledge of the mechanism of NBF transmission to vegetative progeny is thus required to optimize propagation success as well as to determine which plant tissue should be sampled to identify the underlying genetic/epigenetic control. This knowledge will allow development of better molecular-based diagnostics and possibly rehabilitation strategies. Results also challenge the general assumption that age-related disorders in perennial crops are the consequence of an ongoing accumulation of genetic damage in SAMs [3,7–10,32]. Thomas [29]) recently warned that aging in the context of plant biology would be better understood as changes that occur with time, and therefore will embrace the time-based processes of growth and differentiation including maturity, local and regional senescence as well as eventually, mortality. A fundamental yet unresolved problem remains the identification of the plant tissue(s) that act as the repository for age-related "memory" in long-lived perennial plants and plant-clones.

**Author Contributions:** Conceptualization, T.M.G.; methodology, T.M.G. and K.A.S.; validation, T.M.G. and K.A.S.; formal analysis, T.M.G. and K.A.S.; investigation, T.M.G. and K.A.S.; resources, T.M.G.; data curation, K.A.S.; writing—original draft preparation, T.M.G.; writing—review and editing, T.M.G. and K.A.S.; visualization, T.M.G. and K.A.S.; supervision, T.M.G. and K.A.S.; project administration, T.M.G. and K.A.S.; funding acquisition, T.M.G. All authors have read and agreed to the published version of the manuscript.

**Funding:** This research received no external funding.

**Acknowledgments:** The authors wish to acknowledge the support of the California Tree Nut Crop Industry Advisory Board and Almond Board of California for portions of this research.

**Conflicts of Interest:** The authors declare that the research was conducted in the absence of any commercial or financial relationships that could be construed as a potential conflict of interest.

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
