# Peer review of "Propagation of an Epigenetic Age-Related Disorder in Almond Is Governed by Vegetative Bud Ontogeny Rather Than Chimera-Type Cell Lineage"

_horticulturae, doi:10.3390/horticulturae7070190_

Round 1
Reviewer 1 Report
This is an interesting study which reanalyzes the Kester et al. (2004) results in order to study differences in final NBF expression of trees budded with axillary buds from different positions within the budstick, to test of the chimera model for vegetative transmission of epigenetic mosaics in perennial plants. Thus, a large number (2180) of trees propagated from buds of known position within susceptible budstick sources were evaluated
for this disorder.
The manuscript is well written with a very sufficient literature review which provide the necessary background and also, correlation of results with previous references. The final section (summary) is a combination of conclusions, discussion and proposals for future challenges with the use of references and I believe that this should change in title and content.
Bellow are my few remarks:
Fig. 3. in my file the photos were with very low resolution.
page 4 line 3. It will be useful to ad as supplemental material the environmental data for the period of the experiments
4. summary we have abstract in the beginning and summary here. On the other hand, this isn't either conclusions. Maybe it could be conclusions-Future challenges or something like that
Ioannis et al is Filippis et al 2013 please correct it in summary and in References
Author Response
The manuscript is well written with a very sufficient literature review which provide the necessary background and also, correlation of results with previous references. The final section (summary) is a combination of conclusions, discussion and proposals for future challenges with the use of references and I believe that this should change in title and content.
Below are my few remarks:
Fig. 3. in my file the photos were with very low resolution. Higher resolution image added.
page 4 line 3. It will be useful to ad as supplemental material the environmental data for the period of the experiments. To Editor, I could add temperature and rainfall data if desired but this is a 7 year study so it will be a very large file.
- summary we have abstract in the beginning and summary here. On the other hand, this isn't either conclusions. Maybe it could be conclusions-Future challenges or something like that. To Editor, I can change as desired though the author instructions indicated that Horticulturae preferred to do the final formatting.
Ioannis et al is Filippis et al 2013 please correct it in summary and in References. Done.
Reviewer 2 Report
Article
Propagation of an epigenetic age-related disorder in almond is governed by vegetative bud ontogeny rather than chimera-type cell lineage
The article talks about the anatomical and physiological transformations of propagated almonds leading to physiological disorder occurring in the nurseries. The authors concluded that methylation and telomere-length shortening are the cause of epigenetic changes. The work is interesting and brings cognitive qualities to explain the causes of physiological disorder. The large population of plants tested makes the research reliable. Interesting work and suitable for printing in horticulture journal.
Below are suggestions for changes to the text:
Introduction:
Second page, line 15 is: clones in cherry → clones of cherry
Line 18: almond → almonds
Line 5 above the figure 1: refer → refers
Line 3 above the figure 1: in order to → to
M&M
Page 4, Line 3: Please precise location, country, state and coordinates
Line 4: temperature → temperatures
Line 6-7: the scale should be developed whole and clarified, perhaps some graphical explanation is possible since this is the only measurement, its explanation should not raise doubts
Results and discussion
Page 4: vigorous growth on young → growth of young
Figure 4: Why is average BF score? Should be consequently NBF
Below table 1: - controll → control
Author Response
Introduction:
Second page, line 15 is: clones in cherry → clones of cherry Done.
Line 18: almond → almonds Done.
Line 5 above the figure 1: refer → refers Done.
Line 3 above the figure 1: in order to → to Done.
M&M
Page 4, Line 3: Please precise location, country, state and coordinates Added. (see revised text and below).
Line 4: temperature → temperatures Done.
Line 6-7: the scale should be developed whole and clarified, perhaps some graphical explanation is possible since this is the only measurement, its explanation should not raise doubts, Done. (see revised text and below)
These vegetative progeny trees were then planted in replicated blocks and grown under in a commercial orchard conditions in Lost Hills, California in an area Kern Co. in the west-ern San Joaquin Valley (35.6894N, -119.7555W) with where high summer temperatures that favored the expression of the NBF symptoms. NBF in individual progeny trees was rated as previously described in Kester et al. (2004) using visual ratings based on the occurrence of failed buds on current season growth: 0-no NBF, 1-NBF expression limited to 1 tree scaffold, 2-NBF present on more than 1 but less than half of total tree scaffolds, , 3-NBF present on the majority of scaffolds, 4-NBF present on all scaffolds, to and 5-severe NBF with present on all trees scaffolds affected. The entire block of 2180 individual trees were rated annually in March over a seven-year test period by at least three trained evaluators, who reached consensus on each rating.
2.1. Statistical methods.
Results and discussion
Page 4: vigorous growth on young → growth of young Done.
Figure 4: Why is average BF score? Should be consequently NBF Correct-Fixed.
Below table 1: - controll → control Done.
Reviewer 3 Report
The manuscript reports morphological analysis on almond trees from different propagating sources, in order to evaluate symptoms of noninfectious bud failure (NBF). The methodology of the study is extremely simple, and the results, although very simple, are relevant. My main concern with the MS is the inclusion of the theme of genetic and epigenetic variation (which is possibly associated with the expression of NBF) without performing any type of analysis related to it. The result of this inclusion is an extremely speculative discussion and not based on the results obtained. At the summary of the work, once again, the intention of embedding the theme of (epi)genetic variations in a study that did not evaluate this theme is noticeable. Therefore, I do not consider the article apt to be published in Horticulturae.
Author Response
I apologize for the confusion. The intent of this manuscript was not to evaluate genetic versus epigenetic control but rather to examine the pattern of NBF expression in vegetative progeny trees when propagated from sequential axillary buds of the same shoot. Specifically, the objective was to determine whether the expression pattern of axillary buds within the same shoot (as determined by subsequent NBF expression in individual vegetatively propagated trees) was consistent with the sectorial patterning expected using the standard shoot apical meristem cell fate model employed for genetic disorders. The resulting propagation pattern, however, was shown to be very different in this NBF epigenetic-like disorder. Support for an epigenetic rather than genetic control of NBF was presented in earlier (cited) literature. I have rewritten portions of the manuscript, particularly the final section of the introduction to hopefully clarify this confusion.
The revised section is as follows, with newly introduced discussions in bold. (The added references are also included at the bottom of this note). A preprint of more recent research where we utilized a discordant monozygotic twin-based study to further support an epigenetic-like versus genetic control for NBF can be found at: https://www.biorxiv.org/content/10.1101/2021.02.08.430330v1
NBF has been shown to be transmitted to both vegetative propagules and sexual progeny without a pathogenic origin or epidemiological pattern of dispersion (Fenton et al., 1988). Epigenetic-like rather than genetic control has been demonstrated by the age-dependent change in phenotypic expression within the same genotype (clone) (Kester and Gradziel 1996; Kester et al. 2004; Gradziel and Fresnedo-Ramírez 2019; Gradziel et al. 2019). Epigenetic mechanisms, including DNA methylation, have been implicated in phenotypic variation in several plant species (He et al. 2011; Elhamamsy 2016;) including almond (Fernández i Martí et al., 2014; Fresnedo-Ramírez et al., 2017). Recently, Fresnedo-Ramírez et al. (2017) have demonstrated that that NBF-exhibition is associated with DNA methylation and chronological age in almond.
NBF remains a serious threat to the continued production of the dominant cultivar Nonpareil which is currently planted on more than 0.6 million acres, and has resulted in dramatic reductions in plantings of cv. Carmel, which was the second most important commercial cultivar in the early 2000's before being afflicted by NBF (Kester et al 2004; Gradziel and Fresnedo-Ramírez 2019). Because NBF is a clone-age related disorder, control in Nonpareil has been achieved through the careful selection, maintenance and propagation of low clone-age propagation sources (Gradziel et al. 2019). Similar propagation source analysis for the cultivar Carmel demonstrated an increase in NBF expression with increased age of clone propagation-sources (Kester et al. 2004). Results showed that careful selection of budwood source used for tree propagation provided the most effective control of subsequent NBF expression in ensuing commercially grown trees. Trees propagated from the original Carmel breeder’s seedling tree showed the lowest proportion of NBF with increased NBF observed with increased generations of propagation (where generation refers to the number of propagation cycles from the original seedling tree). Propagation methods to minimize NBF expression in commercial tree clones is thus similar to bud-propagation practices to avoid undesirable genetic chimeras. However, the relative importance of individual propagation bud-source within the NBF donor shoot has not previously been evaluated. This study reanalyzes the Kester et al. (2004) results in order to study differences in final NBF expression of trees budded with axillary buds from different positions within the budstick, to test of the chimera model for vegetative transmission of epigenetic-like mosaics in perennial plants.
New references.
Elhamamsy AR. 2016. DNA methylation dynamics in plants and mammals: overview of regulation and dysregulation. Cell Biochemistry and Function 34: 289–298.
He XJ, Chen T, Zhu JK. 2011. Regulation and function of DNA methylation in plants and animals. Cell Research 21: 442–465.
Fenton CAL, Kuniyuki AH, Kester DE. 1988. Search for a viroid etiology for noninfectious bud failure in almond. HortScience 23: 1050–1053.
Fernández i Martí A, Gradziel TM, Socias i Company R. 2014. Methylation of the Sf locus in almond is associated with S-RNase loss of function. Plant Molecular Biology 86: 681–689.
Round 2
Reviewer 1 Report
The authors has incorporate in the manuscript all the suggestions. Thus, I think that the manuscript could be accepted for publication in present form.
Reviewer 3 Report
I reiterate my previous opinion on the manuscript. The modifications presented by the authors are not enough, in my opinion, to make the article publishable. The bias in the introduction and discussion remains the same, and the results presented still do not support the argumentative line. Therefore, I do not recommend publishing this article.
